# Tumor Cell Glycolysis—At the Crossroad of Epithelial–Mesenchymal Transition and Autophagy

**DOI:** 10.3390/cells11061041

**Published:** 2022-03-18

**Authors:** Fabrizio Marcucci, Cristiano Rumio

**Affiliations:** Department of Pharmacological and Biomolecular Sciences, University of Milan, Via Trentacoste 2, 20134 Milan, Italy; cristiano.rumio@unimi.it

**Keywords:** EMT, autophagy, glycolysis, starvation, AMPK, mTOR

## Abstract

Upregulation of glycolysis, induction of epithelial–mesenchymal transition (EMT) and macroautophagy (hereafter autophagy), are phenotypic changes that occur in tumor cells, in response to similar stimuli, either tumor cell-autonomous or from the tumor microenvironment. Available evidence, herein reviewed, suggests that glycolysis can play a causative role in the induction of EMT and autophagy in tumor cells. Thus, glycolysis has been shown to induce EMT and either induce or inhibit autophagy. Glycolysis-induced autophagy occurs both in the presence (glucose starvation) or absence (glucose sufficiency) of metabolic stress. In order to explain these, in part, contradictory experimental observations, we propose that in the presence of stimuli, tumor cells respond by upregulating glycolysis, which will then induce EMT and inhibit autophagy. In the presence of stimuli and glucose starvation, upregulated glycolysis leads to adenosine monophosphate-activated protein kinase (AMPK) activation and autophagy induction. In the presence of stimuli and glucose sufficiency, upregulated glycolytic enzymes (e.g., aldolase or glyceraldehyde 3-phosphate dehydrogenase) or decreased levels of glycolytic metabolites (e.g., dihydroxyacetone phosphate) may mimic a situation of metabolic stress (herein referred to as “pseudostarvation”), leading, directly or indirectly, to AMPK activation and autophagy induction. We also discuss possible mechanisms, whereby glycolysis can induce a mixed mesenchymal/autophagic phenotype in tumor cells. Subsequently, we address unresolved problems in this field and possible therapeutic consequences.

## 1. Introduction

Tumor cells often recur to glycolysis for energy production, even under oxygen-sufficient conditions (the so-called Warburg effect [1]), instead of the more efficient mitochondrial respiration. This phenomenon is referred to as metabolic reprogramming and is now considered one of the hallmarks of cancer [2]. In recent years, however, it has become clear that metabolic reprogramming in tumor cells is not as complete as originally thought. First of all, it has been found that some blood or solid tumors rely mainly on oxidative phosphorylation (OXPHOS) for energy generation [3]. Second, tumor cells relying on aerobic glycolysis for energy production may coexist with tumor cells relying on mitochondrial respiration [4]. In fact, the latter cells can use glycolysis-derived lactate for energy generation by fueling it into the tricarboxylic acid (TCA) cycle and OXPHOS [5]. Third, when glucose concentrations are low, tumor cells can switch back to OXPHOS from aerobic glycolysis [6]. Eventually, individual tumor cells may also exist in a hybrid metabolic state, with aerobic glycolysis and OXPHOS coexisting in these cells [7].

Tumor cell glycolysis, whenever it occurs, is closely linked to two phenotypic changes that affect tumor cells, epithelial–mesenchymal transition (EMT) and macroautophagy (hereafter referred to as autophagy). While both EMT and autophagy are forms of resistance towards tumor cell-autonomous- or tumor microenvironment (TME)-derived stimuli [8], they represent two different responses to these threats. Thus, as a result of EMT, tumor cells acquire several functionalities, such as increased motility, invasiveness, propensity to metastasize, tumor-propagating and immunosuppressive potential and resistance to apoptosis and genotoxic stress [9,10,11,12]. The transition from an epithelial to a mesenchymal phenotype, however, is not an all-or-nothing event. Rather, the transition covers a continuum of intermediate, hybrid phenotypes and it appears as if these hybrid phenotypes, that encompass both epithelial as well as mesenchymal traits, are those that endow tumor cells with the most malignant functionalities [9]. Autophagy, on the other hand, endows tumor cells with the capacity to survive by catabolically degrading cellular material in lysosomes and generating energy from the metabolites deriving from such degradation [13,14].

One crucial signaling pathway for EMT induction is the phosphatidylinositol 3-kinase (PI3K)/AKT (protein kinase B, PKB)/mechanistic target of rapamycin (mTOR) pathway. A detailed description of the events leading to the activation of mTOR and from activated mTOR to EMT can be found in several comprehensive reviews e.g., [9,15,16]. Briefly, the PI3K/AKT/mTOR pathway can be activated by growth factors or other extracellular mediators [16], as well as nutrients, such as certain amino acids, cholesterol or glucose [15,17,18,19,20,21]. The role played by growth factors and extracellular mediators, on one hand, and nutrients, on the other hand, however, is different. Nutrients promote the recruitment of an mTOR-containing protein complex, mTOR complex 1 (mTORC1), to the lysosomal surface [15]. Here, mTORC1 interacts with components of a large molecular complex that includes the Ras homolog enriched in brain (Rheb), which is activated in response to growth factors or extracellular mediators and which, in turn, activates mTOR [22]. This chain of events conforms to a two-signal system, where the first signal (nutrients) plays a permissive role [15], while the second signal (growth factors and extracellular mediators) plays an effector role and is responsible for the eventual activation of mTOR [16]. In addition to PI3K/AKT/mTOR, other pathways (e.g., RAS-RAF-MEK-extracellular signal-regulated kinase (ERK), small mother against decapentaplegic (SMAD), Wnt, Notch, Hedgehog, signal transducer and activator of transcription 3 (STAT3), nuclear factor kappa-light-chain-enhancer of activated B cells (NF-κB)), can induce EMT in tumor cells [10]. These pathways can cross-talk and cooperate with PI3K/AKT/mTOR in EMT induction [9]. PI3K/AKT/mTOR and the other signaling pathways induce the expression of EMT-promoting transcription factors (TF) (e.g., zinc finger E-box binding homeobox 1 (ZEB1), SNAIL, TWIST), which are responsible for the eventual acquisition of mesenchymal traits by tumor cells [9].

For autophagy induction, the key signaling node is adenosine monophosphate-activated protein kinase (AMPK). Conditions of nutrient deprivation and low cellular energy lead to an elevation of the cellular AMP:adenosine diphosphate (ADP):ATP (hereafter AMP:ATP) ratio and, subsequently, to AMPK activation through its phosphorylation, inhibition of its dephosphorylation and allosteric activation [23,24]. AMPK, however, can also be activated under conditions that do not lead to alteration of the AMP:ATP ratio, such as acute glucose deprivation [25], lysosomal damage [26], calcium flux [27], mechanical stress [28], certain cytokines [29], reactive oxygen species (ROS) [30], ionizing radiation and certain chemotherapeutics [31]. Activated AMPK, in turn, activates the autophagy preinitiation complex through phosphorylation of the unc-51-like kinase 1 (ULK1) [13]. The preinitiation complex then phosphorylates components in the autophagy initiation complex, which then converts phosphoinositides in the endoplasmic reticulum to phosphatidylinositol 3-phosphate (PI3P). PIP3 then recruits the so-called ligation machinery, which ligates microtubule-associated protein 1A/1B light chain 3 (LC3) to phosphatidylethanolamine (PE), to form LC3-PE. LC3-PE and components of the ligation machinery promote the formation of double-membrane sequestering vesicles, called autophagosomes. Autophagosomes then fuse to lysosomes, giving rise to autolysosomes, which, subsequently, digest the autophagosome content.

Importantly, AMPK and mTORC1 negatively regulate each other: AMPK inhibits mTORC1 and mTORC1 inhibits AMPK [32,33]. Moreover, AMPK and mTORC1 negatively regulate the activity of each other also at the level of ULK1 activation. Thus, AMPK activates ULK1 by phosphorylating it at six different sites, while mTORC1 inhibits ULK1 by phosphorylating it at a single site, different from those phosphorylated by AMPK [34,35]. This reciprocal, negative regulation underscores the crucial and opposite roles that AMPK and mTOR play in (tumor) cell biology, in conditions of nutrient starvation and nutrient sufficiency, respectively: mTOR activates ATP-consuming anabolic pathways, while AMPK activates ATP-generating catabolic pathways [15,36].

In addition to AMPK-dependent modes, also AMPK-independent modes of autophagy induction have been reported. Thus, autophagy has been shown to be induced in an apparently AMPK-independent manner, in response to genotoxic stress [37], hypoxia [38] or acidic pH [39].

In the following, we discuss the links between glycolysis, EMT and autophagy in tumor cells. We also propose a model aimed at putting in a coherent framework, findings which, at least in part, appear conflicting and contradictory. Beforehand, we would like to stress that, due to obvious space limitations, we reference in this article only part of the many articles that have been published on this topic. For this reason, we apologize to all of the authors whose work could not be cited here. Moreover, in this article we describe activities of components of the glycolytic pathway, in particular glycolytic enzymes, which are unrelated to their role in glycolysis and are usually referred to as non-metabolic activities. Nevertheless, for the sake of simplicity, we refer here to glycolysis to also include such non-metabolic activities, although we recognize that this is not entirely accurate.

## 2. From Glycolysis to EMT or Autophagy

### 2.1. Glycolysis-Inducing Stimuli in Tumor Cells

Many of the stimuli that have been found to induce EMT or autophagy, i.e., tumor cell-autonomous stimuli or stimuli from the TME [8,10,40], have also been reported to induce glycolysis in tumor cells. Thus, for example, stimuli from the TME, such as antitumor therapeutics [41], ROS [42,43], hypoxia [44,45], compressive stress [46], bacterial infection [47], NaCl, H_2_O_2_, ultraviolet radiation and anisomycin [48], high glucose concentrations [49,50,51] or different types of oncogenic drivers [52,53,54], are stimuli that have all been shown to upregulate glycolytic components and glycolysis. Moreover, cell-autonomous stimuli and stimuli from the TME have been shown to cooperate in shaping the glycolytic phenotype of tumor cells [52,54]. Overall, glycolysis, EMT and autophagy are induced in tumor cells by a largely overlapping set of stimuli, suggesting a close functional relationship between the three events.

This knowledge, and the fact that, as we will see in more details in the following, glycolysis can induce either EMT or autophagy, or even a mixed phenotype, encompassing both mesenchymal, as well as autophagic traits, is surprising. In fact, while EMT and autophagy represent two responses to tumor cell-autonomous stimuli or stimuli from the TME, they endow tumor cells with very different functionalities, in order to thwart these stimuli. In the case of EMT, tumor cells respond by displaying an aggressive potential, allowing them to disseminate and metastasize, curb antitumor immune responses and resist apoptosis. In the case of autophagy, tumor cells respond in a defensive manner, entering a state of mere survival until the threat has passed [8,9].

### 2.2. Glycolysis-Induced EMT in Tumor Cells

Glycolysis-induced EMT in tumor cells occurs when the expression and/or activity of components of the glycolytic pathway (transporters, enzymes or metabolites) is increased. Figure 1 gives a schematic representation of glycolytic metabolism, as well as some metabolic pathways that branch off or that modulate steps in the glycolytic metabolism.

In many cases, glycolysis-induced EMT is the result of a redistribution of such components to cellular locations other than the cytoplasm (e.g., the nucleus) and, in the case of glycolytic enzymes, also of activities unrelated to their role in glycolysis. As will be seen in the following, these conditions are essentially the same for glycolysis-induced autophagy or inhibition.

Increased expression and/or activity of glycolytic components that induce EMT has been shown, for example, for the glucose transporter (GLUT) 1 [55], for the fructose transporter GLUT5 [56], for one or more isoforms of various glycolytic enzymes, such as hexokinase (HK) I, HKII [57], HKIII [58], phosphofructokinase 1 (PFK1) [59], phosphoglucose isomerase (PGI) [60,61], aldolase, fructose-bisphosphate (ALDO) A [62], ALDOB [63], phosphoglycerate kinase 1 (PGK1) [64,65], enolase 1 (ENO1) [66], pyruvate kinase M2 (PKM2) [67], lactate dehydrogenase (LDH) A [68], LDHC, [69], or for enzymes involved in the positive regulation of glycolysis, such as 6-phosphofructo-2-kinase/fructose-2,6-biphosphatase (PFK2) isoform 3 or 4 (PFKFB3 or 4) or pyruvate dehydrogenase kinase 1 (PDK1) [45,70,71]. While in most cases, overexpression and/or increased activity of glycolytic components is the consequence of increased transcription and/or translation e.g., [63], in some other cases it can be the consequence of reduced degradation e.g., [72], of posttranslational modifications that enhance the stability of glycolytic enzymes e.g., [73,74,75,76] or of enzymes that positively regulate glycolysis e.g., [77,78]. Eventually, increased activity of a glycolytic enzyme can also be the result of the suppression of an inhibitory interaction [79].

Increased expression and/or activity of individual glycolytic components leads to an overall upregulation of glycolytic metabolism and, under anaerobiosis or in presence of aerobic glycolysis (the so-called Warburg effect), to overproduction of lactate [80]. This, in turn, leads to the generation of an acidic TME. In fact, lactate and an acidic TME have also been shown to promote tumor cell EMT [81,82,83] or EMT-related functionalities [79,84].

Mechanistically, glycolytic components induce tumor cell EMT by different means. One way is through redistribution of glycolytic enzymes, from the cytosol to the nucleus, and interaction with EMT-promoting transcription factors (TF) [67,85,86]; through upregulation of the expression of EMT-inducing master TFs, such as ZEB1 or SNAIL [87,88,89]; by repressing the transcription of *CDH1*, which encodes the prototypic epithelial marker E-cadherin [90].

Glycolytic enzymes can also increase the activity of TFs that induce the expression of glycolytic enzymes e.g., (BTB and CNC homology 1 (BACH1), c-Myc, hypoxia-inducible factor (HIF)-1α, cyclic AMP response element-binding protein (CREB)). This leads to an amplification of glycolysis overall and, consequently, of its EMT-inducing potential [91,92,93,94,95,96]. Incidentally, this also explains why the upregulation of an individual enzyme can give rise to an overall enhancement of glycolytic metabolism [95].

As regards the nuclear localization of glycolytic enzymes, a very interesting case has been described by Liang et al. [90]. They found that PGK1 that had redistributed to the nucleus induced EMT, via repression of E-cadherin expression, through binding to the core promoter region of *CDH1*, while cytoplasmic PGK1 supported tumor cell proliferation through its metabolic function [90]. This shows that a glycolytic enzyme induces EMT upon nuclear localization, while the same enzyme continues to carry out its normal enzymatic task in the cytosol.

Glycolytic enzymes, however, can induce EMT or EMT-related functionalities, not only by redistributing to the nucleus, but also to mitochondria [97]. Thus, in colorectal carcinoma (CRC) cells harboring the BRAF^V600E^ mutation, phosphorylated PDK1 was found to inhibit the pyruvate dehydrogenase (PDH) complex or to induce mitochondrial fragmentation and, by so doing, to promote a glycolytic phenotype, clonogenic potential and metastatic advantage to tumor cells [70].

Moreover, glycolytic enzymes can also induce EMT by interacting and activating components of the signaling pathways (see Section 1) that promote tumor cell EMT, e.g., [67,98,99,100]. Interactions leading to activating posttranslational modifications (e.g., phosphorylation) can also occur with other molecules (e.g., coactivators of TFs) that contribute to EMT induction [101].

While a large body of evidence shows that glycolysis can induce EMT, there is also some evidence suggesting that glycolysis can inhibit EMT. PDK4, for example, has been shown to divert glucose to the TCA cycle in tumor cells undergoing EMT [102]. PDK4 overexpression partially blocked transforming growth factor (TGF)-β-induced EMT, while PDK4 inhibition was sufficient to drive EMT. It was suggested, however, that this might be related to some unique functions of this PDK isoform. Overall, the evidence showing that glycolysis can also inhibit EMT, at least directly, appears to be very scarce. This knowledge does not exclude, however, that EMT might be inhibited indirectly, as a consequence of glycolysis-induced AMPK activation and mTORC1 inhibition, according to the reciprocal, negative regulation between AMPK and mTORC1 that has been discussed in Section 1.

### 2.3. Glycolysis-Induced Autophagy in Tumor Cells

Autophagy can be induced by glycolysis in tumor cells in response to stimuli similar to those that induce EMT. Examples of glycolytic components that are involved in autophagy induction are enzymes, such as HKII [103,104,105,106,107,108], PFK1, platelet isoform (PFKP) [109], glyceraldehyde 3-phosphate dehydrogenase (GAPDH) [43,110], PGK1 [111], PKM2 [112,113], LDHA [114], LDHB [115,116], PFKFP4 [117], PFKFB3 [118], the glucose-regulated protein (GRP) 78 [119], monocarboxylate transporter 1 (MCT1) [120], lactate and acidosis of the TME [121].

Similarly, as we have discussed for EMT induction, glycolytic enzymes that are involved in autophagy induction are overexpressed and/or undergo posttranslational modifications, e.g., [106], interactions with other molecules, e.g., [104,114], or are redistributed to cellular locations other than the cytosol, e.g., [116]. Again, in most cases, autophagy is induced as a result of non-metabolic activities of the enzymes, e.g., [108].

As regards the redistribution of a glycolytic enzyme involved in autophagy induction, an interesting article [122], reporting results in non-transformed cells (mouse embryonic fibroblasts), showed that under glucose, but not amino acid starvation, cytoplasmic GAPDH was phosphorylated on Ser^122^ by activated AMPK. This caused GAPDH to redistribute to the nucleus, where it interacted with and activated the deacetylase sirtuin 1 (SIRT1). Activated SIRT1 then deacetylated and activated several key components of the autophagy pathway. These results suggest that under low glucose, but not low amino acid conditions, a glycolytic enzyme may undergo a discrete modification that promotes autophagy. Interestingly, this modification was performed by AMPK, the main signaling hub responsible for autophagy induction (see Section 1). Nuclear redistribution of GAPDH, as a necessary step for autophagy induction in tumor cells through different mechanisms, has also been reported by others [43,110].

In many, but as we will see, not all cases, a distinguishing feature of glycolysis-induced autophagy in tumor cells is that autophagy is induced in the presence of glucose starvation, e.g., [103,105,106,108,109,119,123,124]. In principle, this may not appear surprising, since nutrient starvation leads to an increase in the AMP:ATP ratio and, consequently, to AMPK activation [36]. Activated AMPK, in turn, is a major autophagy inducer [36]. However, given that in all cases that we have just referenced, an active contribution of glycolytic component(s) was necessary, an increase in the AMP:ATP ratio could not be the cause, or at least not the only cause, of AMPK activation and autophagy induction. In support of this latter possibility, it has been shown that low glucose activates lysosome-associated AMPK through an AMP-independent pathway [125].

The consequences of tumor cell stimulation, in the presence of glucose sufficiency or after glucose depletion, has been investigated in a recent article [102]. Here, the effects of vitamin K on bladder cancer cells were reported. Stimulation with vitamin K, in the presence of glucose sufficiency, led to activation of PI3K/AKT and HIF-1α and, consequently, to upregulation of glycolytic enzymes, induction of glycolysis and inhibition of the TCA cycle. After glucose depletion, upregulation of glycolytic enzymes, together with AMPK activation, mTORC1 inhibition and induction of autophagy were observed.

Autophagy induction, however, has also been observed in the presence of glucose sufficiency, e.g., [104,107,117,118]. In one case, inhibition of glucose uptake inhibited PFKFB3-mediated autophagy induction in renal cancer cells [126], suggesting that an active glycolytic metabolism was necessary for autophagy induction.

Finally, glycolysis has also been shown to induce autophagy without AMPK activation, in some cases by directly acting on components of the autophagy pathway. Thus, HKII has been shown to interact and inhibit mTOR, the main AMPK inhibitor, thereby leading to AMPK activation and autophagy induction [104]. LDHA was reported to induce autophagy by associating with Beclin-1 in tamoxifen-resistant breast cancer cells [114]. PGK1 was acetylated in response to glutamine deprivation and hypoxia and phosphorylated Beclin 1, thereby inducing autophagy and the development of U87 malignant glioma [111].

### 2.4. Glycolysis-Induced Autophagy Inhibition in Tumor Cells

In the previous section, we addressed glycolysis-induced autophagy. Now, we will discuss the opposite, i.e., glycolysis-induced autophagy inhibition. It is very surprising that the glycolytic components that have been reported as being involved in autophagy inhibition are, by and large, the same that induce autophagy. Thus, one or more isoforms of different glycolytic enzymes, such as HKII [127,128,129], PFKFB3 [130,131,132] and PFKFB4 [117,133], PKM2 [100,134,135], and LDHA [136,137], as well as lactate [138], have been reported to inhibit autophagy.

Further, in this case, different mechanisms of action are at work, and the effects are often due to non-metabolic activities of glycolytic enzymes. Thus, in some cases, autophagy inhibition was the consequence of a direct interaction of a glycolytic enzyme with components of the autophagy pathway (e.g., the PFKFB3 with the ubiquitin-associated (UBA) domain of p62/sequestosome-1) [130]. In another case, a glycolytic enzyme (e.g., PKM2) led to autophagy inhibition indirectly, through activation of the PI3K/AKT/mTOR pathway [130,139]. In this and other cases, a reciprocal, negative regulation between EMT and autophagy was observed [136], probably reflecting the reciprocal, negative regulation between mTORC1 and AMPK, as discussed in Section 1.

### 2.5. Tumor Cell Glycolysis and the Mixed, Mesenchymal/Autophagic Phenotype

So far, we have discussed the role of glycolysis in inducing either EMT or autophagy, as if these two phenotypes were mutually exclusive. As already mentioned, this assumption is supported by observations showing that glycolysis-induced EMT and autophagy negatively regulate each other at different levels [136,140,141]. Yet there is substantial literature showing that a mesenchymal and an autophagic phenotype may coexist, or be induced sequentially, in the same tumor cells, in response to similar stimuli, e.g., [8,142,143,144,145,146,147,148]. Only a few studies, however, have investigated whether glycolysis may be involved in the induction of both EMT and autophagy in the same tumor cells. One study showed that starvation enhanced glycolysis and induced both autophagy and EMT in oral squamous carcinoma cells [146]. PFKP was responsible for these effects and for the mixed mesenchymal/autophagic phenotype. The same group also showed that hypoxia caused overexpression of HKII and this led to enhancement of glycolysis and both autophagy as well as EMT, in tongue squamous carcinoma cells [140].

## 3. The Relationship between Glycolysis, EMT and Autophagy—A Tentative Model

In previous sections we have discussed that glycolytic components (transporters, enzymes or metabolites) can induce or inhibit tumor cell EMT or autophagy. The evidence in favor of glycolysis inhibiting EMT is still scarce and will not be discussed further. In the following, we propose a mechanistic model, aimed at accommodating the, in part, contradictory effects that we have discussed in the previous sections.

As a first mainstay of this model, we propose that, in the presence of glucose sufficiency and stimuli (cell-autonomous or from the TME), tumor cells respond by increasing the expression and/or activity of individual glycolytic components, which then induce EMT in tumor cells (Figure 2). Thus, we suggest that upregulation of glycolysis induces EMT as the default pathway. In fact, landscape analyses have suggested that tumor cells first change their metabolism (i.e., from OXPHOS to (aerobic) glycolysis) and then activate EMT [149]. These results have also been confirmed in an experimental setting [150]. Moreover, it has been shown that the highest ^18^F-fluoro-2-deoxy-D-glucose (^18^FDG) accumulation occurs in tumor cells undergoing EMT in hypoxic regions, suggesting an overlapping between EMT and glycolysis [151]. As discussed in Section 2.1, a multitude of different glycolytic components and signaling pathways can be involved in EMT induction. This witnesses a great plasticity, a plasticity that also applies to other glycolysis-induced effects that will be discussed further.

Secondly, we also suggest that, in the presence of glucose sufficiency, EMT induction in tumor cells is accompanied by autophagy inhibition (Figure 3). Again, there is considerable experimental evidence in support of this, e.g., [136,140,141]. In fact, as already discussed in Section 1, one of the main signaling pathways leading to EMT is the PI3K/AKT/mTOR pathway, which, once activated, leads, in addition to EMT induction, to inhibition of AMPK and autophagy, e.g., [32,33]. The PI3K/AKT/mTOR pathway, however, can also inhibit autophagy by acting downstream of AMPK, i.e., on components of the autophagy pathway itself, e.g., [34,35,152].

Overall, we propose that, in the presence of glucose sufficiency and non-metabolic stimuli, glycolysis leads to autophagy inhibition as part of the default pathway that leads to EMT induction.

The picture changes greatly when tumor cells are faced with the combination of glucose starvation and non-metabolic stimuli (Figure 4). We suggest that, in this case, the combination of low glucose and increased expression and/or activity of glycolytic components, leads to AMPK activation and autophagy induction. AMPK activation may not necessarily occur as a result of an increased AMP:ATP ratio. In fact, it has been shown that, in the presence of low glucose, the glycolytic enzyme ALDO is not occupied by fructose 1,6-bisphosphate (F1,6BP), and this lack of occupancy promoted the formation of lysosomal complexes that led to AMPK activation [25]. More recently, another glycolytic metabolite, dihydroxyacetone phosphate (DHAP), has been shown to signal glucose availability to mTORC1 [153]. This suggests that decreased levels of DHAP, due to glucose starvation, may lead to mTORC1 inhibition and, consequently, to AMPK activation and autophagy induction. Similarly, earlier work had shown that under low glucose conditions, the lack of occupancy of GAPDH, by its substrate glyceraldehyde 3-phosphate (G3P), leads to mTORC1 inhibition, through association with an mTORC1-interacting protein (Rheb) [154]. Further, in this case, mTORC1 inhibition is expected to lead to AMPK activation and autophagy induction. Of note, mTORC1 inhibition may also be the consequence of AMPK activation (see Section 1). Therefore, glycolysis-induced activation of AMPK, under the conditions described in this section, may, per se, induce and reinforce mTORC1 inhibition, promoted by decreased levels of DHAP or unoccupied GAPDH.

The mechanism whereby increased levels of unoccupied enzymes or decreased levels of metabolites lead to AMPK activation may also help explain how autophagy can be induced in the presence of glucose sufficiency. Thus, as a logical extension of the observation that a lack of occupancy of ALDO by F1,6BP leads to AMPK activation [25], stimulus-induced upregulation of ALDO may give rise to an increased fraction of unoccupied enzymes, leading to AMPK activation and autophagy induction. A similar mechanism may also apply to GAPDH. Here, insufficient occupancy of upregulated GAPDH [154] by G3P, even in conditions of glucose sufficiency, may lead to mTORC1 inhibition and, indirectly, to AMPK activation. We refer to this mechanism of AMPK activation as “pseudostarvation”.

We have mentioned that, in response to acute glucose deprivation, upregulated glycolytic enzymes can induce AMPK activation and, possibly, autophagy, in the presence of an unaltered AMP:ATP ratio. Later phases of glucose deprivation, or when glucose deprivation is accompanied by lack of other nutrients (e.g., glutamine) until severe nutrient stress ensues, have been shown to lead to progressive increases in the AMP:ATP ratio [25], with AMP activating cytosolic AMPK after lysosomal AMPK and, subsequently, all pools of AMPK, including mitochondrion-localized AMPK [125]. This suggests a progressive activation of the different cellular pools of AMPK at increasing degrees of metabolic stress [125]. This may reinforce and/or replace those stimuli that lead to AMPK activation and autophagy induction, in the absence of an increased AMP:ATP ratio, during early phases of glucose deprivation. As regards this point, so far, we have discussed only glycolysis-induced autophagy in conditions of glucose sufficiency or glucose deprivation, but we have not addressed the possible involvement of glycolysis in the induction of autophagy, in response to deprivation of other nutrients. In fact, relatively little has been reported on this issue. We have already discussed an article addressing this point [111], which showed the involvement of a glycolytic enzyme in the induction of autophagy in response to glutamine deprivation and hypoxia. Overall, however, the information appears to be too scant to draw any conclusion.

At this point, it remains to be explained how glycolysis can induce either EMT or autophagy, in the presence of nutrient (glucose) sufficiency and non-metabolic stimuli, and how EMT and autophagy can coexist in the same population of tumor cells, i.e., the mixed mesenchymal/autophagic phenotype. We believe that the two issues are closely linked and we will discuss them together (Figure 5). The difficulty in explaining these two outcomes arises from the knowledge that the two main signaling nodes that control the induction of EMT or autophagy, i.e., PI3K/AKT/mTOR and AMPK, are under reciprocal negative control, as already discussed.

One possibility to explain how glycolysis can induce autophagy alone, or a mixed mesenchymal/autophagic phenotype in tumor cells, is to assume that the negative control exerted by mTOR on AMPK and ULK1, or on components of the autophagy pathway itself, becomes disabled. In fact, there are indications that this may occur. Thus, we have already referenced one article showing that HKII interacted and inhibited mTOR and, consequently, led to autophagy induction [104]. More recently, it has been reported that mTORC1 activity is greatly reduced during mitotic arrest [155]. This suggests that during mitotic arrest the negative control of mTORC1 on AMPK may get lost. Moreover, mitotic arrest is accompanied by a decline in the mitochondrial mass and an increase in the AMP:ATP ratio, conditions known to lead to AMPK activation [156]. These are conditions conducive to AMPK activation and this activation may then lead to the induction of autophagy and other AMPK-induced biological activities [24]. Thus, AMPK has been shown to increase glycolytic metabolism by augmenting glucose uptake through an increase in the activity and expression of the transporter GLUT1 [157], or the glycolysis-promoting enzyme PFKFB3 [156]. This latter article showed that that the glycolysis-inducing effect of AMPK took place during prolonged mitotic arrest, suggesting that, under such conditions, AMPK activation may increase glycolytic metabolism. This may then lead to an amplification of AMPK-induced biological activities, including autophagy induction. If these events occur in tumor cells with an epithelial phenotype, then a predominantly autophagic phenotype would arise in these cells. If, on the other hand, they occur in tumor cells having already acquired mesenchymal traits, then they would give rise to a mixed, mesenchymal/autophagic phenotype.

A second possibility derives from the observation that AMPK has been shown to induce EMT in tumor cells by increasing the expression of the EMT master TF Twist1 [158]. Thus, glycolysis-driven AMPK activation may also give rise to a bifurcate phenotypic rewiring of tumor cells, leading both to EMT and autophagy induction, offering another explanation on how a hybrid mesenchymal/autophagic phenotype may occur.

Still another possibility that may help explain how glycolysis can induce autophagy alone or a mixed mesenchymal/autophagic phenotype, derives from the knowledge, discussed before, that glycolysis may induce autophagy by directly acting on autophagic components. This would allow to bypass the requirement for AMPK for autophagy induction and, consequently, also the negative control exerted by mTORC1 on AMPK and autophagy induction. We have also mentioned, however, that the PI3K/AKT/mTOR pathway can inhibit autophagy by acting downstream of AMPK, i.e., on components of the autophagic machinery itself [34,35,152], and this could also lead to inhibition of glycolysis-driven autophagy induction that bypasses the requirement for AMPK activation. To our knowledge, however, there is no formal proof that this may actually occur.

Subsequently, one may also consider the possibility that a given stimulus, or a combination of stimuli, dictates whether upregulated glycolysis induces autophagy or a mixed mesenchymal/autophagic phenotype instead of EMT. In support of this possibility comes the evidence, already extensively discussed before, that the combination of stimuli and low glucose induces an autophagic response, while the combination of stimuli and glucose sufficiency induces either EMT or autophagy. Of note, this mechanism is not mutually exclusive with the ones that we have discussed before. Thus, a stimulus that induces a prolonged mitotic arrest, such as a mitotic inhibitor [159], may lead to upregulation of glycolysis, which, under these conditions, would preferentially lead to AMPK activation and autophagy induction, with or without concomitant mesenchymal traits.

The mixed phenotype discussed here, i.e., the coexistence of mesenchymal and autophagic markers in tumor cells, recalls the coexistence of epithelial and mesenchymal markers in tumor cells, i.e., the hybrid epithelial–mesenchymal phenotype [160,161,162] that we have already discussed in the initial part of this article. The phenotypic and functional relationship between these two mixed tumor cell phenotypes remains to be investigated.

## 4. Conclusions

In this article we have reviewed the effects of glycolysis on tumor cell EMT and autophagy. We have also proposed a model, in order to accommodate, in a coherent mechanistic framework, the large body of, in part, contradictory experimental data that have been published on this topic over the years.

In spite of this effort, there are, however, still many unanswered questions. Here, we will raise just of few of them. First, in this article, we have discussed that glycolysis can induce EMT and induce or inhibit autophagy and how this can occur. At this point, one is also led to ask whether glycolysis-independent pathways exist and what is their relevance compared to the overall economy of these events. We have already referred to several articles suggesting that a close relationship does, indeed, exist, between glycolysis and EMT induction and, according to our model, autophagy inhibition [149,150,151], with evidence showing that tumor cells first undergo a metabolic reprogramming towards glycolysis and, subsequently, undergo an EMT [149]. On the other hand, several articles have reported AMPK-independent modes of autophagy induction, which also appear to be glycolysis-independent [37,38,39]. Overall, we feel that, at present, the most balanced view is to state that glycolysis can play a causative role in the induction of EMT and autophagy, but also that glycolysis-independent modes exist. In these cases, glycolysis, whenever it is upregulated in tumor cells, may play an amplifying role.

Second, we have also proposed that glycolysis can induce autophagy instead of EMT in the presence of glucose sufficiency and stimuli that induce mitotic arrest, a condition that favors AMPK activation and inhibits mTORC1 activity. The question now arises whether other modes of glycolysis-dependent autophagy induction exist. In fact, we have referred to evidence showing that glycolysis can induce autophagy by directly acting on autophagic components [111,114] and that PI3K/AKT/mTOR can inhibit autophagy in the same way [34,35,152]. This suggests that PI3K/AKT/mTOR may exert a negative control on this mode of autophagy induction, and that such a negative control must be disabled, in order for glycolysis-induced autophagy to proceed. Again, in the absence of experimental evidence, this issue remains a matter of speculation.

Third, we have mentioned that EMT reflects, in many cases, a hybrid phenotype, encompassing both epithelial and mesenchymal markers, and it is this hybrid phenotype that is associated with the most malignant traits of tumor cells. It would be very interesting to know whether glycolysis induces a hybrid EMT phenotype or a full-blown mesenchymal phenotype, or both, and if there are other conditions, unrelated to glycolysis, that dictate up to which point the EMT continuum proceeds in response to glycolytic stimuli.

Fourth, in this article we have discussed, almost exclusively, the consequences of glycolysis-induced autophagy due to glucose deprivation and, only marginally, the consequences of deprivation of other nutrients and more severe forms of metabolic stress. This is an important point to investigate in the future, because increasing degrees of metabolic stress have been shown to lead, progressively, to the involvement of all cellular pools of AMPK, starting from lysosomal AMPK [125], with consequences on EMT or autophagy induction that remain to be clarified.

The unambiguous identification of all circumstances whereby glycolysis promotes EMT or autophagy induction is not an issue of mere academic interest, since, as discussed at the beginning of this article, both EMT and autophagy represent important mechanisms of resistance towards antitumor drugs and survival under unfavorable conditions that may arise in the tumor cells themselves and/or in the TME.

This latter point brings up the question of whether the insights that derive from the knowledge discussed in this article may entail new possibilities of pharmacological intervention for tumor therapy, in particular, as regards the possibility to inhibit tumor cell EMT or autophagy through the modulation of glycolytic metabolism. One of the most surprising aspects that we have faced during the preparation of this manuscript is the large number of glycolytic components and pathways that can be activated and used by tumor cells, in order to induce EMT and/or autophagy. This reflects a great plasticity of the system and the possibility to follow different routes in order to achieve similar goals, a property of fundamental importance, in the case that one of these routes becomes obstructed, whether because of pharmacological intervention or tumor-autonomous reasons. Given that, from a pharmacological point of view, one possibility to circumvent this heterogeneity is to target the end results of activated glycolysis in tumor cells, i.e., EMT and/or autophagy [10,163,164,165,166,167]. Alternatively, glycolytic components may become pharmacological targets if tumor-specific alterations that remain constant over time (so-called “metabolic rigidity”) [168] can be identified and detected with suitable biomarkers in patients. In any case, deepening our understanding on the role of glycolysis in the modulation of EMT or autophagy of tumor cells appears as a road to take, in order to gain further insight into the fundamental mechanisms of tumorigenesis.

## Figures and Tables

**Figure 1 cells-11-01041-f001:**
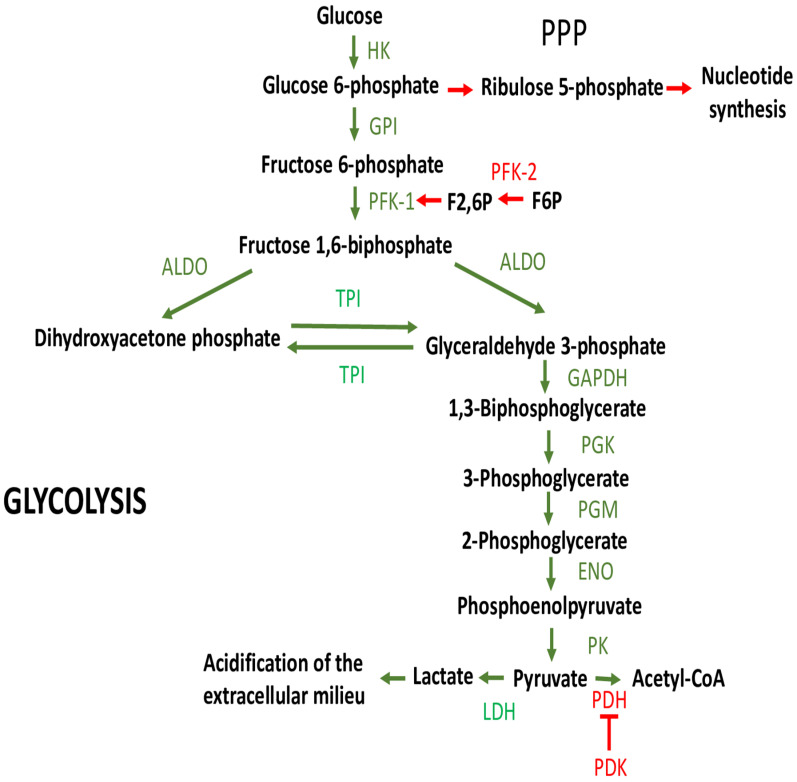
Glycolysis. Glycolytic enzymes and metabolites are depicted as well as some of the pathways that branch off or that modulate steps in the glycolytic metabolism (see text for more details). Green arrows refer to steps that are inherent to glycolysis, red arrows to reactions that branch off or modulate glycolysis. ALDO, aldolase, fructose-bisphosphate; ENO, enolase; GAPDH, glyceraldehyde 3-phosphate dehydrogenase; GPI, glucose 6-phosphate isomerase; HK, hexokinase; LDH, lactate dehydrogenase; PDH, pyruvate dehydrogenase; PDK, pyruvate dehydrogenase kinase; PFK-1, phosphofructokinase-1; PFK-2, phosphofructokinase-2 (including the isoforms 6-phosphofructo-2-kinase/fructose-2,6-biphosphatase 3 (PFKFB3) and PFKFB4); PGK, phosphoglycerate kinase; PGM, phosphoglycerate mutase; PK, pyruvate kinase; PPP, pentose phosphate pathway; TPI, triosephosphate isomerase.

**Figure 2 cells-11-01041-f002:**
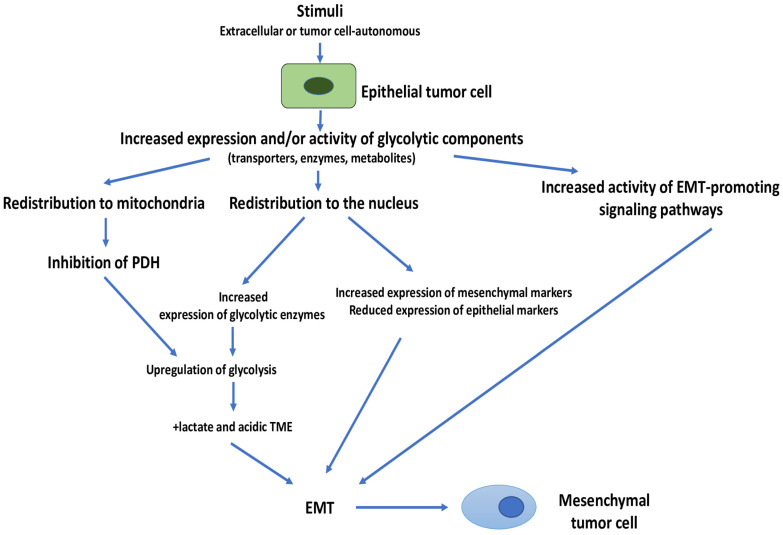
Glycolysis-induced EMT in tumor cells. Stimuli, either tumor cell-autonomous or from the TME (extracellular) increase the expression and/or activity of one or more glycolytic components (transporters, enzymes or metabolites). These components then induce EMT upon redistribution to cellular locations other than the cytosol, or by interacting and increasing the activity of signaling pathways that induce EMT. Overall upregulation of glycolysis is another consequence, and this can lead to enhanced production and secretion of lactate and to the generation of an acidic TME, changes that are also known to induce EMT in tumor cells. EMT, Epithelial–Mesenchymal transition; PDH, pyruvate dehydrogenase; TME, tumor microenvironment.

**Figure 3 cells-11-01041-f003:**
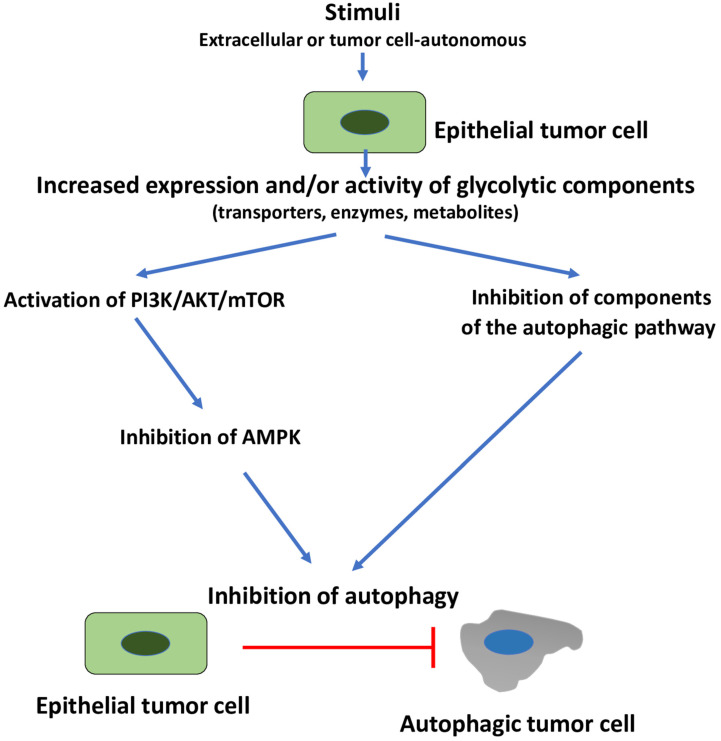
Glycolysis-induced autophagy inhibition. Stimuli, either tumor cell-autonomous or from the TME (extracellular) increase the expression and/or activity of one or more glycolytic components (transporters, enzymes or metabolites). This can lead to activation of several signaling pathways. One of these pathways is the PI3K/AKT/mTOR pathway which leads also to AMPK inhibition and, consequently, to autophagy inhibition. Autophagy inhibition can occur also due to an inhibitory interaction of a glycolytic enzyme with components of the autophagy pathway. AMPK, adenosine monophosphate-activated protein kinase; EMT, epithelial–mesenchymal transition; PI3K/AKT/mTOR, phosphatidylinositol 3-kinase/AKT/mechanistic target of rapamycin.

**Figure 4 cells-11-01041-f004:**
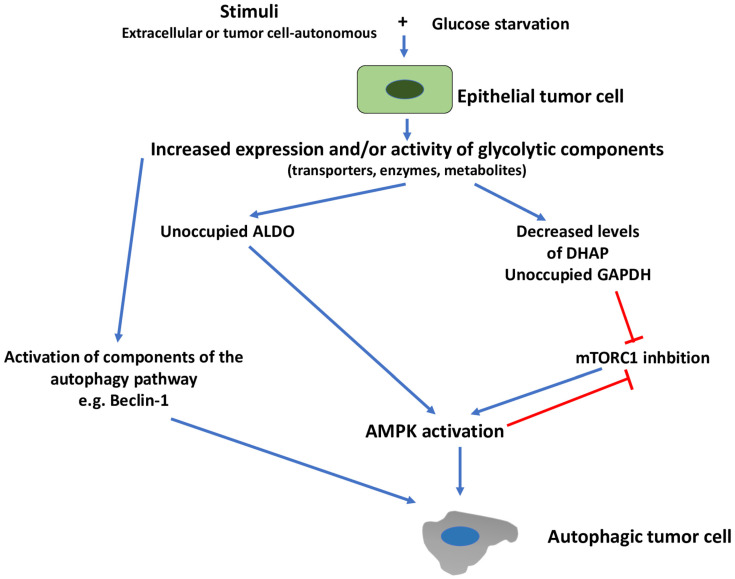
Glycolysis-induced autophagy in the presence of glucose starvation. Stimuli, either tumor cell-autonomous or from the TME (extracellular) increase the expression and/or activity of one or more glycolytic components (transporters, enzymes or metabolites). In the presence of glucose deprivation, however, unoccupied glycolytic enzymes (e.g., ALDO, GAPDH) or decreased levels of glycolytic metabolites (e.g., DHAP) lead to AMPK activation through various mechanisms (see text for details). ALDO, aldolase, fructose-bisphosphate; AMPK, adenosine monophosphate-activated protein kinase; DHAP, dihydroxyacetone phosphate; GAPDH, glyceraldehyde 3-phosphate dehydrogenase; mTORC1, mechanistic target of rapamycin complex 1.

**Figure 5 cells-11-01041-f005:**
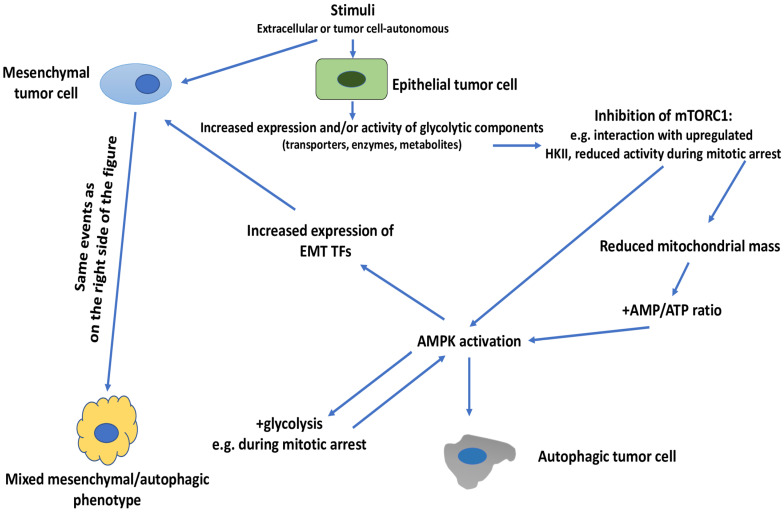
Glycolysis-induced autophagy or mixed, mesenchymal/autophagic phenotype in the presence of glucose sufficiency. Stimuli, either tumor cell-autonomous or from the TME (extracellular) increase the expression and/or activity of one or more glycolytic components (transporters, enzymes or metabolites). At least one glycolytic component (HKII) has been shown to interact with mTOR and inhibit mTORC1. Inhibition of mTORC1 may also occur during mitotic arrest. This inhibition then leads to AMPK activation and autophagy induction. Prolonged mitotic arrest may also promote AMPK-driven upregulation of glycolysis which, through mechanisms similar to those described before (e.g., upregulation of HKII), may lead to AMPK activation and autophagy induction. When these events are induced in epithelial tumor cells, they give rise to an autophagic tumor cell. When they are induced in a tumor cell that has already acquired mesenchymal traits, then they are expected to give rise to a mixed, mesenchymal/autophagic phenotype. AMPK, adenosine monophosphate-activated protein kinase; ATP, adenosine triphosphate; EMT, epithelial–mesenchymal transition; HKII, hexokinase II; mTOR, mechanistic target of rapamycin; mTORC1, mTOR complex 1; TF, transcription factor.

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
