# Peer review of "Tumor Cell Glycolysis—At the Crossroad of Epithelial–Mesenchymal Transition and Autophagy"

_cells, 2022, doi:10.3390/cells11061041_

Round 1
Reviewer 1 Report
The manuscript by Fabrizio Marcucci and Cristiano Rumio entitled “Tumor Cell Glycolysis–at the Crossroad of Epithelial-Mesenchymal Transition and Autophagy” reviews the current state of glycolysis and cell plasticity.
This is a nicely presented, exhaustive and critical review, rarely seen nowadays.
I have one major criticism and some minor ones.
My major criticism is that the authors refer to EMT as a one or the other state, rather than plasticity and the “middle” stages. In the last few years, partial or hybrid EM has been highlighted as the stage where metastasis take place rather than acquired, via EMT, M state.
This plasticity is also important in the many signalling pathways described, where partial hypoxia and partial glycolysis, described in the MS, also play roles. In Biology nothing is black and white, rather the 5,000 shades of grey.
A short explanation for pEMT or hybE/M would be nice for the reader.
Minor issues:
Pp178. or by acting as transcriptional repressor of the prototypic epithelial marker E-cadherin [90]
It should place the gene name (CDH1) as the regulation is at that level
Pp187. described by Liang C et al [90].
It is nice to use either full name/surname or only the surname and et al.
The term “brain tumorigenesis“, although used, is too sloppy. The authors should refer the the proper tumour type as there is no such thing as (whole) brain morigenesis
Figure 2. It says “Mesenchymal umor cell” – missing “t”. Same in Figure 5.
Author Response
We have addressed the points raised by the reviewer as follows:
We have now added a definition of hybrid EMT in the revised version (see lines 51-55). It should be noted, however, that we had briefly addressed this issue also in the previous version of the manuscript (see lines 916-917 of the current version). We did so, however, in the very final part of the manuscript and it is certainly useful to have a definition of this concept also in the initial part of the manuscript. In the conclusions we have now added some sentences on the interest to investigate whether glycolysis induces a hybrid EMT or a full-blown mesenchymal phenotype (lines 949-954 of the present version).
The acronym of the gene (CDH1) has now been entered at line 230 of the present version.
Reference 90 is now indicated after Liang et al at line 239 of the present version.
We have now replaced the term brain tumorigenesis with “with development of U87 malignant gliomas” (lines 377-378 of the present version).
We have now rectified the mistake in Figs. 2 and 5.
Reviewer 2 Report
The presented review by Marcucci and Rumio starts with a nice, simple and yet comprehensive overview of glycolysis, glycolysis-related pathways and their connection/integration to EMT and autophagy.
The work is generally presented on a sound scientific basis using good English language and has now major lacks of information. However, some spell checking and linguistic phrasing corrections are needed to improve the overall quality and eliminate possible misunderstandings.
In section 2, the review is mainly characterized by enumerations of effects published with regard to glycolysis/glycolytic enzymes together with either EMT or autophagy and could be improved by some brief discussions to put known results into context. I understand that both EMT and autophagy are very complex cellular mechanisms and to account for all published data is almost impossible. Nevertheless, the manuscript could highly benefit from brief discussions of known literature and connecting them with each other. Actually, this has very nicely been done in section 3 and section 2 could be similarly good.
Open questions are nicely mentioned both in section 3 and 4 while they are discussed in more detail in the latter.
In section 4, the authors discuss amongst others that the high diversity of effects from the glycolytic machinery on EMT and autophagy make it unlikely that pharmacological interference with metabolic pathways may be beneficial. However, this diversity results mainly because effects here are summarized from multiple tumor entities. Thus, within one tumor type, it may still be possible to target both EMT and autophagy simultaneously and would, in my opinion, be preferable compared to targeting components more downstream in the whole network. This should be also discussed in section 4 as well.
Additionally, figures could be improved by a better focus on the main effects instead of including too many details to depict a better overview at a glance. Moreover, I would suggest to generally revise the figure layout in order to better visualize the effects and not only contain text boxes.
Overall, the review is scientifically correct and comprehensive but could be improved by a revision of the descriptions and the figure layout.
Author Response
The have addressed the points raised by the reviewer as follows:
The first point raised by this reviewer was not very clear to us. In any case, in the present version of the manuscript we have added a new paragraph (lines 150-158 of the present version) where we evidence the relevance of tumor cell EMT and autophagy and where we underscore the difficult link between these two phenotypic changes of tumor cells, which represent very different answers to siilar stimuli and stressors. We hope that this modification meets the point raised by the reviewer.
As regards the second major point we have now added at the end of the manuscript (lines 999-1002) of the present version) some lines that take into account his viewpoint about the possibility of tumor-specific metabolic rearrangements and the possibility to pharmaceutically targeting them.
We have also mad some modifications to the figures as suggested by the reviewer. We have tried to simplify and reduce the text whenever possible. Admittedly, we did not make major modifications because this would not have been possible in the 5 days that we had available to resubmit the modified version of the manuscript.
Reviewer 3 Report
Wanted summary on the topic which is relevant for EMT and cancer related research. Authors describe in detail where and how tumor cell metabolism might intersect with autophagy and EMT process. There are not major issues concerning the text of the manuscript both in terms of its content. Still, there are several aspects of the present form manuscript which should be modified to improve its usefulness for the targeted audience.
These include:
1) authors operate with EMT as with the process which has the definitive form. On the other hand, as based on the current understanding the situation with EMT is more complex since it seems to be rather plastic sequence of events which may not be completed. Thus EMT observed in various tumors may only be partial or somehow started and completed. This fact has to be taken in consideration in particular in the light of plasticity of tumor cells. Authors should at least mention and briefly discuss this stance since otherwise one might be lead to the conclusion that EMT occurrs homogeneously in tumors.
2) Authors use ery complicated sentences which sometimes (more often) fail to deliver the final message. It is suggested authors simplify it. The typical case is the paragraph (page 2, starting with "Importantly, AMPK and mTORC1 regulate each other) which should be rephrased.
Author Response
We have addressed the points raised by this reviewer as follows:
We have now added a definition of hybrid EMT in the revised version (see lines 51-55). It should be noted, however, that we had briefly addressed this issue also in the previous version of the manuscript (see lines 916-917 of the current version). We did so, however, in the very final part of the manuscript and it is certainly useful to have a definition of this concept also in the initial part of the manuscript. In the conclusions we have now added some sentences on the interest to investigate whether glycolysis induces a hybrid EMT or a full-blown mesenchymal phenotype (lines 949-954 of the present version).
We carefully went through the whole manuscript. In fact, we found quite a number of typos that now have been amended. They can be found throughout the manuscript. Moreover, we also tried to simplify a number os sentences including the one indicated by the reviewer.